



# Lidar vertical observation network and data assimilation reveal key processes driving the 3-D dynamic evolution of PM$_{2.5}$ concentrations over the North China Plain

Yan Xiang[1], Tianshu Zhang[2, 1], Chaoqun Ma[3], Lihui Lv[1], Jianguo Liu[2], Wenqing Liu[2, 1], and Yafang Cheng[3]

[1]Institutes of Physical Science and Information Technology, Anhui University, Hefei 230601, China

[2]Key Laboratory of Environmental Optics and Technology, Anhui Institute of Optics and Fine Mechanics, Chinese Academy of Sciences, Hefei 230031, China

[3]Minerva Research Group, Max Planck Institute for Chemistry, Mainz, Germany

**Correspondence**: Yan Xiang (yxiang@ahu.edu.cn) and Yafang Cheng (yafang.cheng@mpic.de)

**Abstract**: China has made great efforts to monitor and control air pollution in the past decade. Comprehensive characterization and understanding of pollutants in three-dimension (3-D) are, however, still lacking. Here, we used data from an observation network consisting of 13 aerosol lidars and more than 1000 ground observation stations, combined with a data assimilation technique, to conduct a comprehensive analysis of an extreme heavy aerosol pollution (HAP) over the North China Plain (NCP) from November–December 2017. During the studied period, the maximum hourly mass concentration of surface PM$_{2.5}$ reached ~390 μg·m$^{-3}$. After assimilation, the correlation between model results and the independent observation sub-dataset was ~50% higher than the that without the assimilation, and the root mean square error was reduced by ~40%. From pollution development to dissipation, we divided the HAP in the NCP (especially in Beijing) into four phases—an early phase (EP), a transport phase (TP), an accumulation phase (AP), and a removal phase (RP). We then analyzed the evolutionary characteristics of PM$_{2.5}$ concentration during different phases on the surface and in 3-D space. We found that the particles were mainly transported from south to north at a height of 1-2 km (during EP and RP) and near the surface (during TP and AP). The amounts of PM$_{2.5}$ advected into Beijing with the maximum transport flux intensity (TFI) were through the pathways in the relative order of the southwest > southeast > east pathways. The dissipation of PM$_{2.5}$ in the RP stage (with negative TFI) was mainly from north to south, with an average transport height of ~1 km above the surface. Our results quantified the multi-dimensional distribution and evolution of PM$_{2.5}$ concentration over the NCP, which may help policymakers develop efficient air pollution control strategies.




## 1 Introduction

Frequent heavy air pollution has exerted significant impacts on air visibility, climate, human health, and other environmental concerns (Gao et al., 2017a; Pokharel et al., 2019; Su et al., 2020). As a developing country with the largest population in the world, China's air quality has exhibited an obvious improvement trend in recent years (Cao et al., 2017; Zhang and Cao, 2015). Regional air pollution in China is still serious, however, especially the heavy aerosol pollution (HAP) caused by fine particulate matter ($PM_{2.5}$) in winter, which has attracted attention worldwide (Cheng et al., 2016; Li et al., 2017b; Zheng et al., 2015; Zheng et al., 2019). Therefore, providing a reliable distribution of the $PM_{2.5}$ concentration of HAP, especially at any time and at any height in a given region, is particularly important in the quest of the public to avoid health problems and to provide government policy makers with help in designing effective controls (Hu et al., 2015).

Compared with other air pollutants (e.g., ozone and nitrogen dioxide), $PM_{2.5}$ has a longer atmospheric lifetime (3–5 days), during which it can be transported vertically to great heights and horizontally hundreds of kilometers (Wang et al., 2017; Zhang et al., 2014), depending on the meteorological conditions (e.g., relative humidity and precipitation) and chemical composition (Yang et al., 2017). Previous study demonstrated that regional transport plays an important role for pollution formation in major cities of China, e.g., transport contributes over 50% of the $PM_{2.5}$ mass concentration in Beijing city, Shanghai city, Hangzhou city, Guangzhou city, Hong Kong and Chengdu city during the relatively polluted period (Sun et al., 2017). From 2005–2010, annually, about 35.5% (32.8 $\mu g \cdot m^{-3}$) of the $PM_{2.5}$ in Beijing was attributed to regional transport from the North China Plain (NCP), within which up to 60.4% (64.3 $\mu g \cdot m^{-3}$) from southerly and westerly air flows (Wang et al., 2015). Since the 2013 implementation of the most stringent clean air policy in China, the control of local pollution sources has led to the rapid reduction of total $PM_{2.5}$ concentration (Zhang et al., 2019c). It should be noted, however, that the local contributions, intra-regional transport, and inter-regional transport accounted for 47% (12.7 $\mu g \cdot m^{-3}$), 25% (6.6 $\mu g \cdot m^{-3}$), and 28% (7.6 $\mu g \cdot m^{-3}$), respectively, of the total $PM_{2.5}$ for the Beijing-Tianjin-Hebei (BTH) region from 2014–2017, with the 2017 contribution of regional transport to the BTH concentration rate ranging from 32.5–68.4% (Dong et al., 2020).

Previous studies have shown that it is difficult to use surface observations to characterize the impact of upper-level pollutants in the atmosphere (Huang et al., 2018b), which is affected by local emissions, regional transport, meteorological conditions, geographical factors etc.



(Tao et al., 2020). Therefore, understanding the key processes that drive the dynamic temporal
and spatial evolutionary characteristics of pollutants on the NCP is essential for revealing the
source and transport of aerosols, which has different radiative forcing at different heights
(Kumar et al., 2017). Actually, stereo-monitoring devices and technologies, such as lidar (Chen
et al., 2019b; Fan et al., 2019; Sheng et al., 2019), MAX-DOAS (Hong et al., 2018; Zhang et
al., 2020), and satellite remote sensing (Pang et al., 2018; Schwartz et al., 2012; Zhang et al.,
2019a), can reveal the vertical distribution of pollutants at different heights (Heese et al., 2017;
Tian et al., 2017). Due to the limited spatial and temporal observations, however, it is
impossible to provide physical and chemical properties in the atmosphere at any time period
and on any path, which makes it difficult to directly reveal the formation and source of pollution.
On the other hand, although the distribution of pollutants can be simulated by air quality
models (Huang et al., 2018a; Zhang et al., 2008), large uncertainties remain, mainly from the
influence of emission inventory, meteorological fields, and some hypothetical conditions
(Chen et al., 2017; Huang et al., 2016; Xu et al., 2016). Fortunately, the above observed data
and the results of the model can be fused using data assimilation techniques, which can correct
the model simulation results via the observed data (Ma et al., 2019; Wang et al., 2013).
Research has shown that mainstream data assimilation (DA) technologies, including 3DVAR
(Jiang et al., 2013; Ma et al., 2018), 4DVAR (Yumimoto et al., 2008), and EnKF (Chen et al.,
2019a), can be used to assimilate observation data from the surface, remote sensing data (such
as AOD) from satellites, and vertical profile data from lidar, all of which can be used to improve
the performance of the model, including the simulation of $PM_{2.5}$ and $PM_{10}$.
In this study, we analyzed the observation data from a vertical observation network
consisting of 13 lidars and surface observation stations during an extreme pollution event in
eastern China, especially in the NCP. Next, all of the data were utilized by the Gridpoint
Statistical Interpolation (GSI) three-dimensional (3-D) variational (3DVAR) data assimilation
system to revise the $PM_{2.5}$ results from the WRF-Chem simulation (Pagowski et al., 2014).
Finally, the multi-dimensional evolutionary characteristics of $PM_{2.5}$ at the surface and in the
vertical layer, as well as the 3-D distribution, were analyzed in detail. Although data
assimilation has been applied in China using surface observation network data (Gao et al.,
2017b), AOD (Liu et al., 2011; Saide et al., 2013; Saide et al., 2014; Schwartz et al., 2012),
and lidar data (Cheng et al., 2019), to our knowledge, this is the first attempt in China to apply
lidar network data to assimilation technology, from which the high-precision 3-D distribution
of pollutants can be provided, thus supplying effective data support for clarifying the formation
mechanism of pollutants (Zheng et al., 2017).



## 2 Measurements and methods

### 2.1 Lidar observation network

The vertical aerosol observation network of the NCP was composed of 13 aerosol lidar monitoring stations (Fig. 1), covering four main transport channels of Beijing pollutants, including the southwestern transport path of Baoding City (BD), Shijiazhuang City (SJZ), Xingtai City (XT), Handan City (HD), Xinxiang City (XX), and Yangquan City (YQ); the southern transport path of Dezhou City (DZ) and Jining City (JN); the southeastern transport path of Langfang City (LF), Cangzhou City (CZ), and Zibo City (ZB); the eastern transport path of Tianjin City (TJ); and a lidar in the urban area of Beijing (BJ).

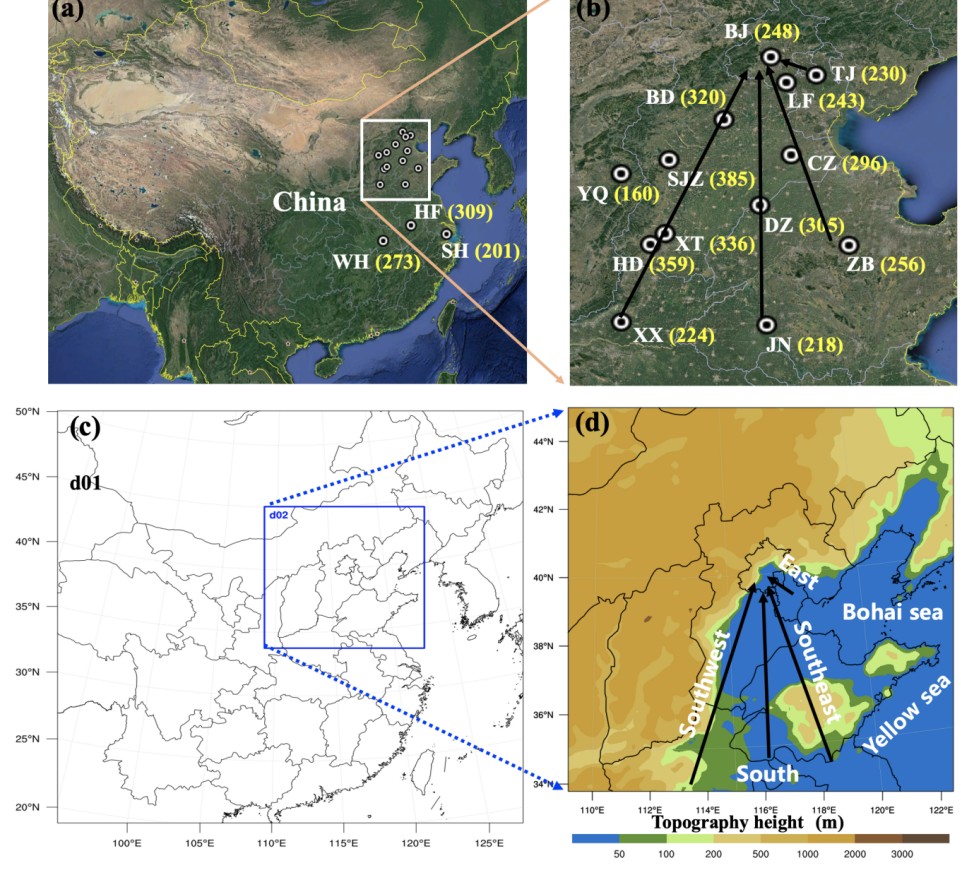

**Figure 1.** © Google maps of (a) China with the studied cities and (b) the North China Plain with all the lidar stations. The data in brackets are the maximum PM$_{2.5}$ concentrations ($\mu g \cdot m^{-3}$) at the surface during the observation period. (c) Two-nested WRF-Chem modeling domains and (d) topographic elevation data in d02. The black arrows in (b, d) from left to right show that the main pollution pathways of Beijing come from the four directions of southwest, south, southeast, and east.



The lidar system was developed by the Anhui Institute of Optics and Fine Mechanics
(AIOFM), Chinese Academy of Sciences (CAS), and was used for the long-term continuous
observation of aerosol vertical distribution. The lidar system adopted the Nd: YAG laser, which
emits a 532-nm wavelength, with 30-mJ single-pulse energy and 10–30-Hz pulse repetition
frequency. The vertical resolution is 7.5 m, with a time resolution of 3–10 min. The detection
blind area is 0.1 km; more specific technical details can be found in other literature (Xiang et
al., 2019). The vertical distribution of the aerosol extinction coefficient was retrieved using the
Fernald method (Fernald, 1984), which is more suitable for vertical detection and more
accurate than the Collis (Collis et al., 1964) and Klett (Klett, 1981) methods (Lu et al., 2015).
Furthermore, combining the extinction coefficient with the $PM_{2.5}$ *in-situ* surface observations,
the vertical distribution of the $PM_{2.5}$ mass concentration in the boundary layer was obtained
using the empirical formula fitting method, which has proven to be reliable and highly accurate;
the specific technical details can be found in other literature (Lv et al., 2017a; Lv et al., 2017b;
Tao et al., 2016). In addition, an image recognition algorithm was used to evaluate the height
of the atmospheric boundary layer (Barrera et al., 2019; Xiang et al., 2019).
**2.2   WRF-Chem model configurations**
The WRF-Chem chemical transport model was used to investigate the particulate
concentrations and meteorological parameters in the study area and was configured with nested
domains consisting of $100 \times 100$ (36 km) and $103 \times 103$ (12 km) grids (Figs. 1c and 1d). The
domain had 41 vertical layers from the surface to 50 hPa. To better simulate the conditions
within the boundary layer, the resolution of the boundary layer was increased, and 20 layers
were set in the range of 0–2 km. The initial and boundary meteorological conditions were
derived from the 6-h National Centers for Environmental Prediction Final Analysis data with
$1° \times 1°$ spatial resolution. The inventory of anthropogenic emissions for 2016 was obtained
from the Multi-resolution Emission Inventory for China (MEIC) data with $0.25° \times 0.25°$
resolution (Zhou et al., 2017). Terrestrial biogenic emissions were estimated using the Model
of Emissions of Gases and Aerosols from Nature (MEGAN) model (Chatani et al., 2011). The
gas-phase chemistry module CBM-Z and the Model for Simulating Aerosol Interactions and
Chemistry (MOSAIC) aerosol module were used in this simulation. Detailed information
concerning the model configuration is provided in Table S1. The model runs from November
20, 2017–December 9, 2017, and the results from November 25–December 9, 2017 were used
for the analysis in Section 3.
**2.3   GSI 3DVAR DA system**





The GSI DA (Gridpoint Statistical Interpolation Data Assimilation) system provides
3DVAR analysis by minimizing the cost function as shown below (Gao et al., 2017b):
$$J(x) = (x - x_b)^T B^{-1}(x - x_b) + \left(y - H(x)\right)^T R^{-1}\left(y - H(x)\right) \qquad (1)$$
In this equation, $x$ is the analysis vector, $x_b$ denotes the background vector, $y$ is an observation
vector, $B$ represents the background error covariance matrix, $R$ represents the observation error
covariance matrix, and $H$ is the observation operator used to transform model grid point values
to observed variables, which was performed via interpolation in our research. The background
error covariance matrix was calculated using the National Meteorological Center (NMC)
method (Parrish and Derber, 1992; Saide et al., 2013), which simulated the difference of results
at the same time (November 25, 2017) with two different starting times (November 20, 2017
and November 21, 2017, respectively). The 1-hour assimilated window data included 13
groups (see Fig. 1 for site distribution) of $PM_{2.5}$ vertical profiles retrieved from lidar, and the
surface $PM_{2.5}$ data from hundreds of surface monitoring stations (see Fig. 5 for site distribution)
from the China Environmental Monitoring Center. The observation errors of $PM_{2.5}$ ground and
its vertical distribution (through the ground $PM_{2.5}$ fitting method in Section 2.1) originated
from measurement errors and representative errors. The measurement error was computed
using $\varepsilon_0 = 1.5 + 0.0075 * obs$ (Pagowski et al., 2014), where $obs$ indicates observed values.
The representative error was computed using $\varepsilon_r = \gamma \varepsilon_0 \sqrt{\Delta x / L}$ (Elbern et al., 2007), where $\gamma$ is
the adjustable scale factor (we used the value of 0.5 recommended by the GSI system), $\Delta x$ is
the model grid resolution (we selected 12 km of domain 2), and $L$ is the influencing radius (we
used 60 km).
## 3    Results and discussion
### 3.1    Evaluation of assimilation performance using vertical $PM_{2.5}$ data

24       In order to evaluate the improvement of model simulation performance from data

assimilation using lidar vertical profile data and surface station data, considering the sharp
decline of $PM_{2.5}$ value at 1km height (Fig. 6), only the non-assimilation and assimilation results
at the surface, 0.2 km, 0.5 km, and 1 km were compared, as shown in Fig. 2. These data were
selected from five of the most polluted stations, including the cities of TJ, LF, BD, SJZ, and
XT. It should be noted that these observation data were not assimilated, which means that the
following comparisons are independent (Bocquet et al., 2015). Obviously, the data assimilation
used can greatly improve the simulation accuracy. Compared with the observation data at
different heights, the simulation results of $PM_{2.5}$ levels under the condition of non-assimilation
were higher (Figs. 2 a–d), the root-mean-square error (RMSE) was $52.14 \pm 20.27$ μg·m$^{-3}$, and


the correlation coefficient was only 0.56 ± 0.15. Correspondingly, the results of PM$_{2.5}$
simulated with assimilation were closer to the observed values (Figs. 2 e–h), the RMSE was
33.07 ± 14.69 μg·m$^{-3}$, which represents a reduction of about 40% in simulation error after
assimilation. The correlation coefficient was 0.81 ± 0.10, demonstrating that the simulation
accuracy was improved by about 50% after assimilation.

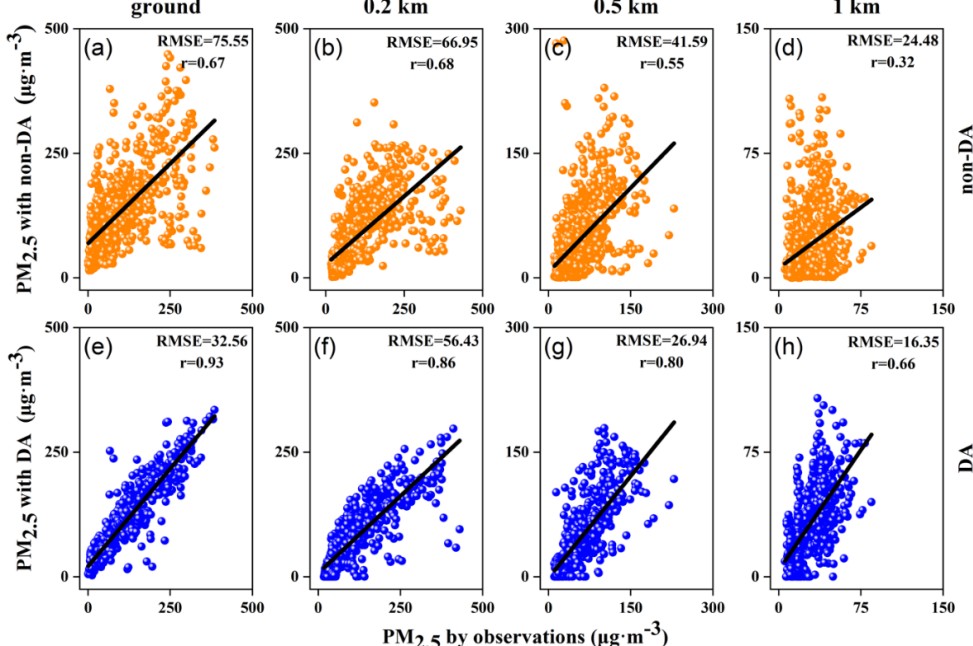

**Figure 2.** PM$_{2.5}$ mass concentration comparison results from lidar at different heights (b–d, f–h) and
surface observations (a, e) with non-assimilation simulations (a–d) and assimilation simulations (e–h).

9         In addition, compared with the simulation with assimilation (Fig. 5 in Section 3.3), the

results without assimilation were significantly higher than the observed values (Fig. S1),
especially during the pollution period (Figs. S1d, S1e), which may be due to the simulation
error caused by the model (Zhang et al., 2016). Meanwhile, the comparison of the three-
dimensional results (Fig. 7 in Section 3.5 and Fig S2) further reveals that the simulation results
of upper air PM$_{2.5}$ may also overestimate the actual values, which demonstrates the importance
of data assimilation in capturing the three-dimensional structure of pollution.
**3.2    The four phases from aerosol pollution development to dissipation**

17         Joint observations and analyses have been widely performed in an effort to reveal the

heavy aerosol pollution (HAP) in the NCP region (Li et al., 2016; Zhang et al., 2018). The key
processes of a HAP event, from aerosol pollution development to dissipation, usually include



an early phase (EP), a transport phase (TP), an accumulation phase (AP), and a removal phase
(RP) (Yuan et al., 2019; Zhong et al., 2017), classifications that are based on the increase and
decrease of PM$_{2.5}$ mass concentration in Beijing (BJ) caused by changes in meteorological
conditions. Here, the curves in Fig. 3 shows the temporal evolution of PM$_{2.5}$ mass concentration
monitored at the surface in different cities on the NCP from November 25–December 9, 2017,
while the superimposed colors represent the time-varying profiles of the simulated wind fields
in BJ, Baoding (BD), Dezhou (DZ), and Langfang (LF), respectively. Overall, PM$_{2.5}$ with high
concentrations was usually associated with pronounced southerly winds (S in Fig. 3) or
southwesterly winds (SW in Fig. 3), while the PM$_{2.5}$ concentrations decreased significantly
under the prevailing northerly winds (including the wind directions of N, NW, and NE in Fig.

11   3).

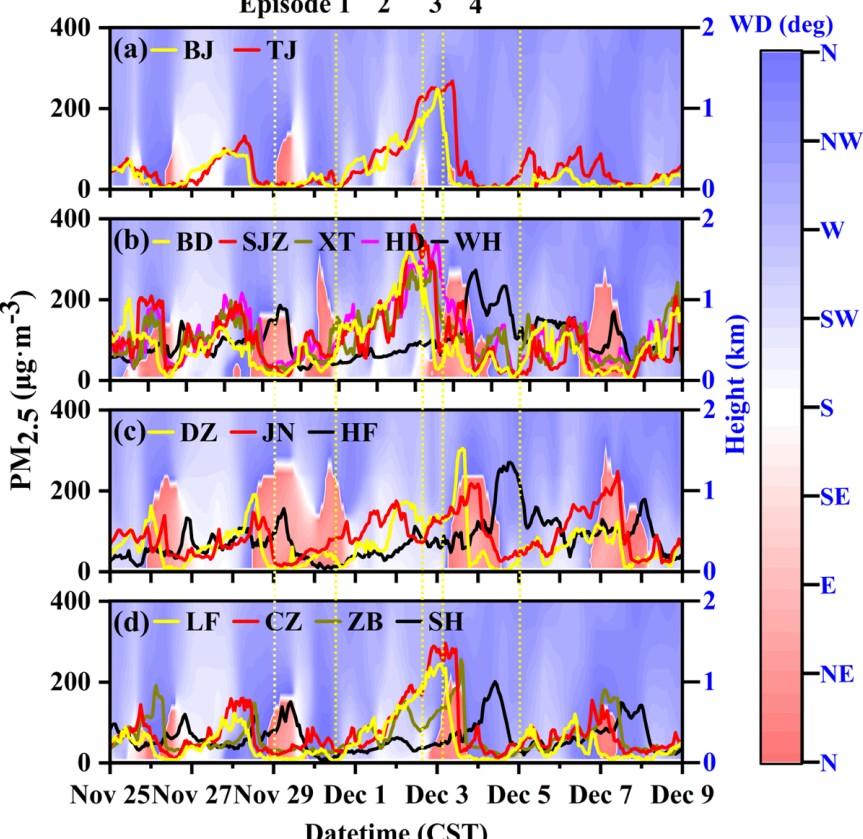

**Figure 3**. Surface PM$_{2.5}$ observations from different cities: (a) Beijing (including Tianjin) and its (b) southwest cities, (c) southeast cities, and (d) east cities for the period November 25–December 9, 2017. Superimposed colors represent the time-varying profiles of the simulated wind fields in Beijing, Baoding, Dezhou, and Langfang, respectively.





Furthermore, in order to characterize the evolution of $PM_{2.5}$ during different pollution
phases, the period from November 29–December 5 was selected as a typical extreme HAP
event covering the four pollution phases. This extreme pollution event lasted more than 4 days
and featured a regional transport process. During the EP (November 29–noon November 30;
episode 1 in Fig. 3), the air quality in BJ and its surrounding areas such as Tianjin (TJ) was
relatively good, with an average $PM_{2.5}$ value of ~15 $\mu g \cdot m^{-3}$, while slight pollution occurred to
the southwest of BJ, including BD, Shijiazhuang (SJZ), Xintai (XT), and Handan (HD), with
an average value of ~50 $\mu g \cdot m^{-3}$.
During the TP (approximately the morning of December 2; episode 2 in Fig. 3), the
variation of $PM_{2.5}$ concentration was more sensitive and responded rapidly to the wind shift
from northerly to southerly, causing the $PM_{2.5}$ concentration in Beijing to increase quickly from
~30 $\mu g \cdot m^{-3}$ to ~150 $\mu g \cdot m^{-3}$, while southwest of Beijing (e.g., BD, SJZ, XT, and HD) the $PM_{2.5}$
concentration increased rapidly to ~200 $\mu g \cdot m^{-3}$. Research has revealed that the pollutant
transport south of Beijing, especially in the southwest areas (the Taihang Mountains), is the
most important contribution source of Beijing pollutants (Zhao et al., 2020). During the AP
(approximately December 3; episode 3 in Fig. 3), diffusion of the pollutants was difficult due
to the occurrence of a surface temperature inversion in Beijing (Fig. 4) (Wang et al., 2019),
which caused the maximum concentration of $PM_{2.5}$ in Beijing to reach ~250 $\mu g \cdot m^{-3}$.
Meanwhile, the $PM_{2.5}$ concentrations in TJ, LF, BD, and SJZ reached maximum values of ~270,
250, 320, and 390 $\mu g \cdot m^{-3}$, respectively. Conversely, the pollution levels in Shanghai (SH),
Hefei (HF), and Wuhan (WH) in the southernmost section of the NCP were relatively low,
with average values < ~60 $\mu g \cdot m^{-3}$.
During the RP (approximately December 5; episode 4 in Fig. 3), the wind direction
shifted from southwest to north, transporting the relatively clean air in the north to the south,
and thereby causing the pollutant concentrations in Beijing to decrease rapidly. In just 9 hours,
the air quality improved from heavy pollution to excellent, and the $PM_{2.5}$ concentrations in the
NCP also decreased significantly. Finally, by noon on December 4, the pollutant concentrations
in the NCP had reached a low level, with an average value of ~40 $\mu g \cdot m^{-3}$. In contrast, due to
the continuous southward advection of pollutants, serious pollution occurred in SH, HF, and
WH, where the $PM_{2.5}$ concentrations reached maximum values of ~210, 310, and 280 $\mu g \cdot m^{-3}$,
respectively. These findings are also consistent with the results of previous studies on the
regional transport of regional pollutants to the Yangtze River Delta (Hua et al., 2015), which
showed them to be due to the continuous southward flow of northwest and northeast winds.

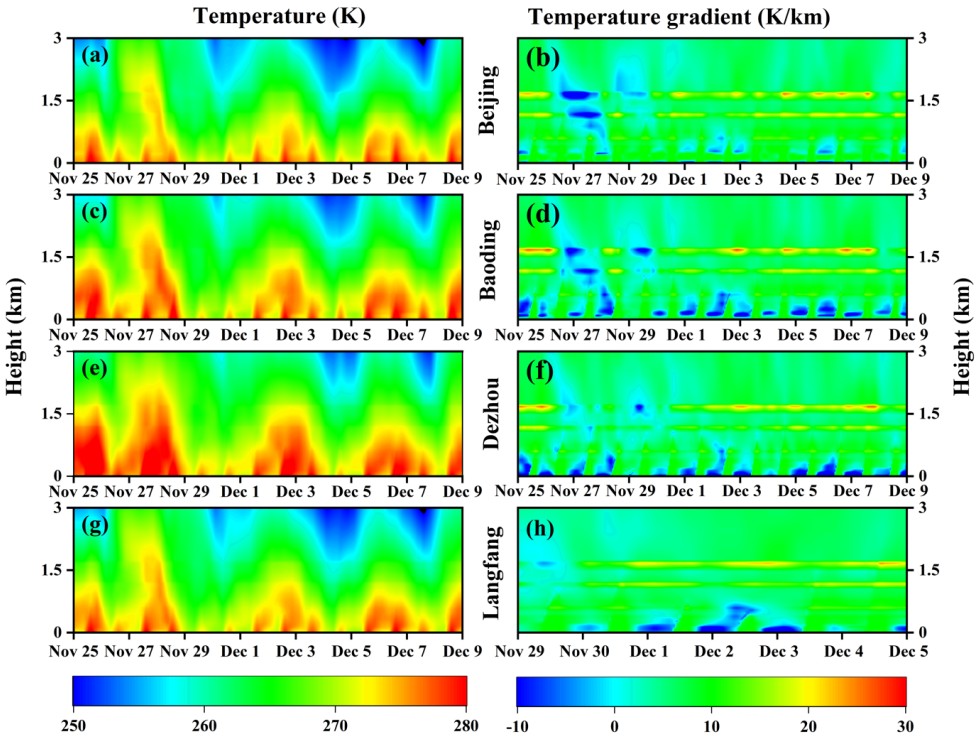

**Figure 4.** Time series of vertical temperatures (a, c, e, g) and temperature gradients (b, d, f, h) from
Beijing (a, b), Baoding (c, d), Dezhou (e, f), and Langfang (g, h) simulated by the WRF-Chem model.
**3.3    Spatial distribution of PM$_{2.5}$ concentration in the surface layer**

5       Additionally, in order to analyze the pollution characteristics of the NCP, the spatial

distribution results of PM$_{2.5}$ after data assimilation were plotted in Fig. 5 for all phases. The
high concentrations of PM$_{2.5}$ in BJ were recorded during the TP, AP, and beginning of the RP,
while the PM$_{2.5}$ concentrations at other times were lower. Moreover, during the EP, only the
eastern cities of Shanxi (SX) Province experienced moderate pollution levels (Fig. 5a). During
the TP, the pollutants in the south-central NCP were transported to the north of the NCP (Figs.
5b and c) as a result of the southwesterly wind field, and under the superposition of the local
pollutant emissions from each city (Li et al., 2017a), the cities on the windward side of the
Taihang Mountains (e.g., HD, SJZ, and BD) quickly developed varying levels of heavy
pollution. In addition, during the AP, due to the large-scale inversion (Figs. 4b, d, f, h) caused
by the rapid temperature rise (Figs. 4a, c, e, g) of the NCP region at upper levels, the
atmospheric stratification was stable, causing the pollutant loading on the NCP (including BJ,
BD, SJZ, HD, LF, CZ, and elsewhere) to increase (Fig. 5d), nearly reaching their pollution
maxima (Fig. 3). Meanwhile, during the RP, affected by the cold air at upper levels (Figs. 4a,





c, e, g) from the northwest and the shift in wind direction over the NCP from southwest to
north, the pollution severity gradually eased from north to south (Fig. 5e), with the air quality
in the northern part of the region improving significantly (Fig. 5f).

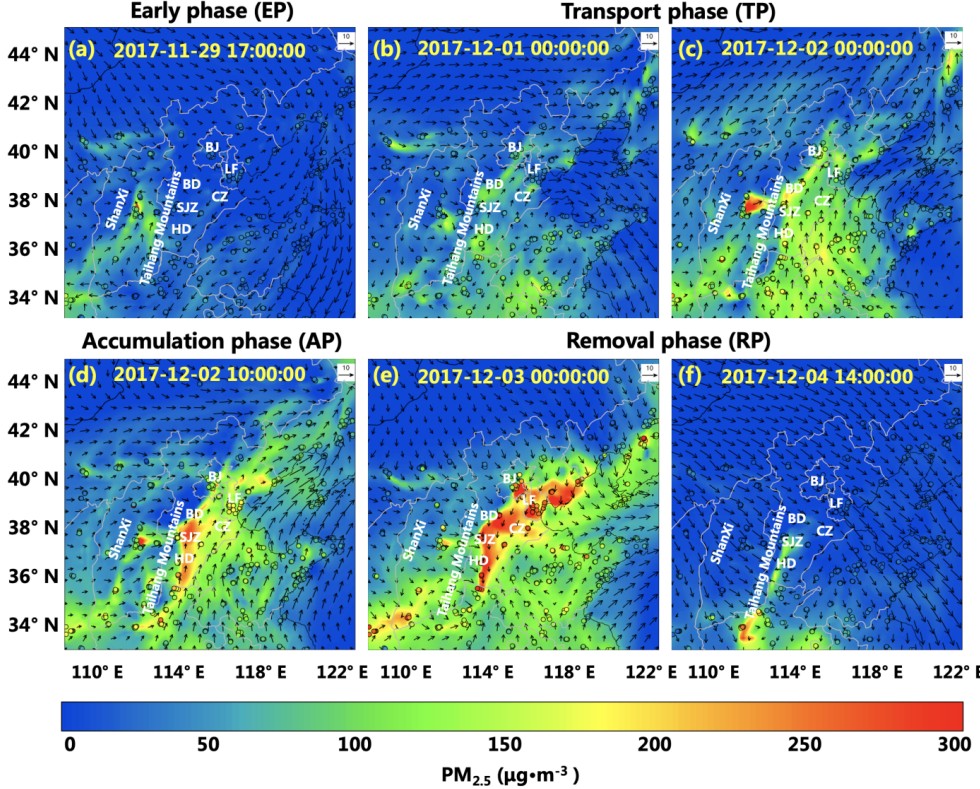

**Figure 5**. Spatial distribution of PM$_{2.5}$ in the surface layer during different phases after assimilation.
The black arrows indicate the wind direction. The circles represent the *in-situ* surface observations.
**3.4    Vertical distribution of aerosols observed by the lidar network**

8         In order to quantify the characteristic vertical distribution of aerosols, the observed

aerosol extinction coefficients from the 13 lidar stations in the NCP were plotted, as shown in
Fig. 6. These results revealed that on November 29, the aerosol concentration at the surface
was relatively low, although pollutant transport at heights of 1–2 km (see Figs. 8a, e) occurred
at six stations (BD, SJZ, YQ, XT, HD, and XX) on the windward side of the Taihang Mountains.
The upper air transport of pollutants continued until December 1, at which it merged with the
surface flow. Contrary to this, the pollutant transport from north to south occurred at a height
of 1 km during the RP (e.g., Figs. 6b, d–g). In addition, the atmospheric boundary layer height
(ABLH) reached its highest value of the observation period from November 29 to 30, averaging
more than 1.5 km. The ABLH began to decrease on December 1, averaging approximately 1


km on that day. The lowest value of the ABLH occurred on December 2–3, when its average
dropped to less than 0.5 km, making it difficult for pollutants to diffuse and causing heavy
pollution in the NCP (Li et al., 2017c). Fortunately, on December 4, the atmospheric boundary
layer gradually lifted, which was conducive to the diffusion of pollutants.

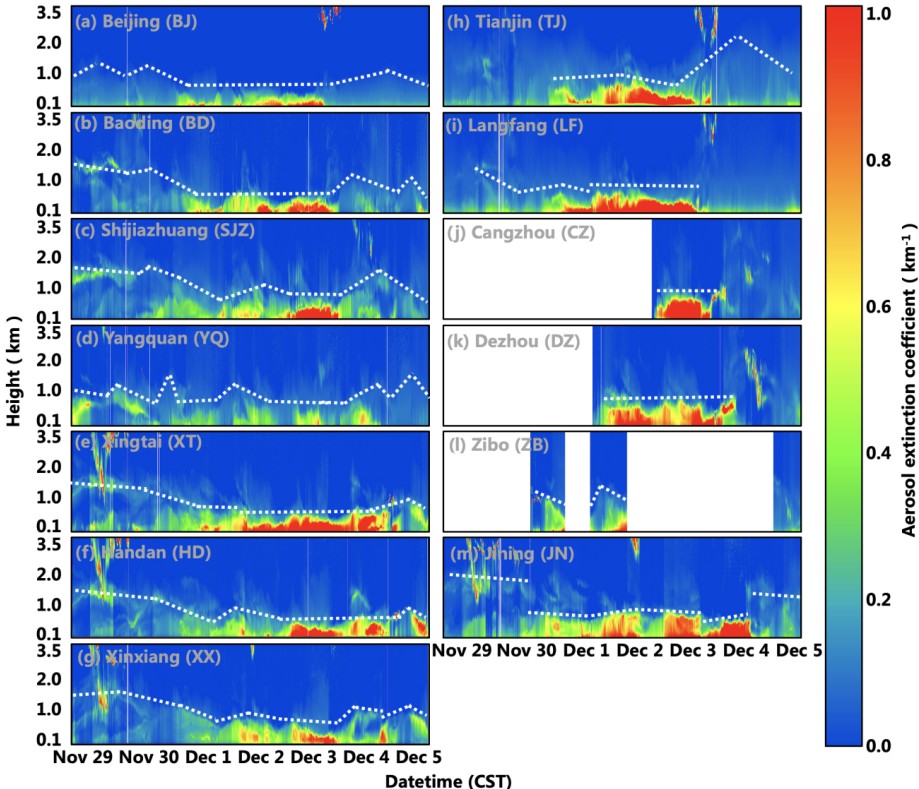

**Figure 6.** Time series of vertical distributions of the aerosol extinction coefficient observed on the
North China Plain from November 29–December 5, 2017. The white dashed lines represent the
approximate atmospheric boundary layer height.
**3.5   Dynamic 3-D evolution of the PM$_{2.5}$ concentrations**
Figure 7 presents the 3-D distribution of PM$_{2.5}$ after assimilation, which clearly shows
the generation, dissipation, transport, and diffusion characteristics of pollutants in the
atmosphere. The tops of the boxes in the figure depict the wind speeds 10 m above the surface.
During the EP, the high-concentration pollutants only occurred in the upper air within ~1 km
of the surface in SX Province (e.g., YQ). During the TP, the high-concentration pollutants were
mainly found on the windward side of the Taihang Mountains (southwest pathway), and the
loading height of PM$_{2.5}$ was < 1 km, which is illustrated in Fig. 8. During the AP, the average
concentration of pollutants > 200 μg·m$^{-3}$ mainly occurred near the surface. Meanwhile, the
pollutants with low concentrations at upper levels could be transported to the Bohai Sea.
During the RP, high-concentration pollutants $> 100$ μg·m$^{-3}$ simultaneously occurred over the
Bohai Sea and the Yellow Sea.

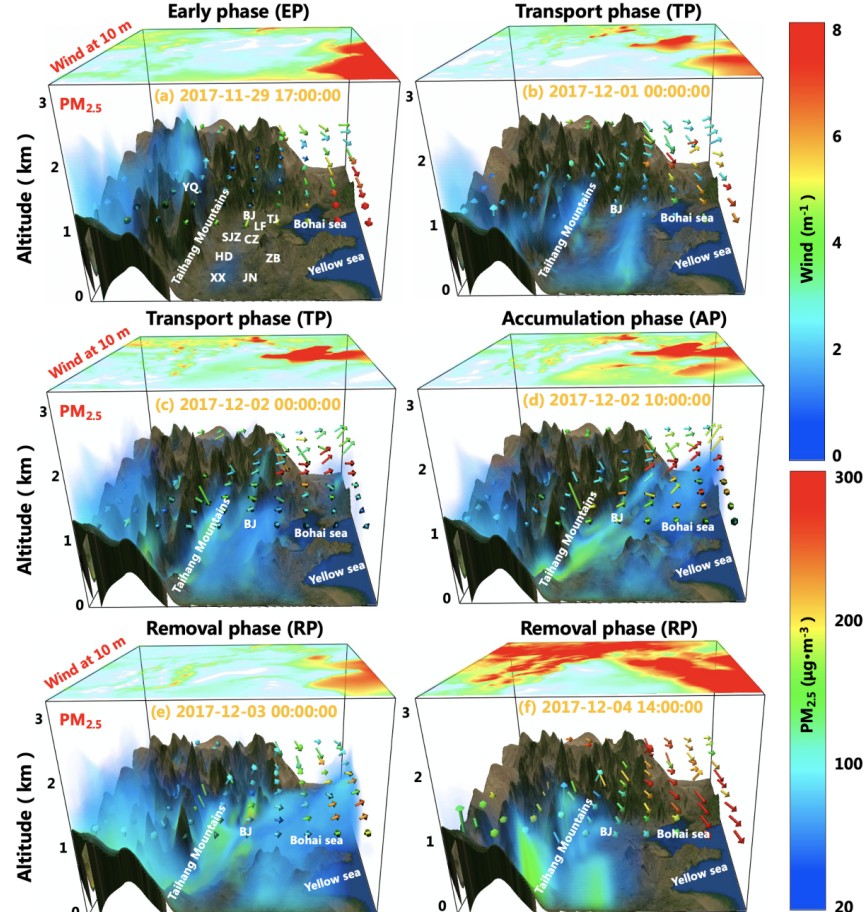

**Figure 7**. Three-dimensional distribution of PM$_{2.5}$ during different phases after assimilation. Colors
within the boxes depict the PM$_{2.5}$ concentrations. The color-coded arrows represent the wind direction
and speed at 1 km. On the tops of the boxes, the spatial distributions of wind speed at 10 m are plotted.
### 3.6 Quantification of regional transport of PM$_{2.5}$
To evaluate the variation of pollutants along different transport pathways at different
stages, we plotted the vertical profile of the PM$_{2.5}$ cross-section along the main pollution
pathways of Beijing come from the four directions of southwest, south, southeast, and east (see
Figs. 1b, d). As shown in Fig.8, at XX and XT (located at the start of the southwest transport
pathway, Fig. 8a), the PM$_{2.5}$ concentration is more than 200 μg·m$^{-3}$ at a height of 1 km (Fig.
8a), and the surface PM$_{2.5}$ concentraiton at JN (located in the south pathway) also exceeds 200





μg·m⁻³ (Fig. 8b). These high concentrations of pollutants were transported to SJZ, BD, LF, BJ,
and other cities via southwest winds (Figs. 8e, f, g). At the same time, vertical downdrafts
reduced the height of loading of aersol layer to ~0.6 km (Fig. 8e). Different from the southern
(including southwest, south, and southeast) transport pathways, the pollutants in TJ were
mainly from BJ outflow in all stages of the eastern transport pathways (Figs. 8d, h, l, p). In
addition, wind direction inconsistencies at the origin (XX, JN, and ZB) and target location
(Beijing) of the transport pathways occurred at the beginning of the removal phase (Figs. 8i–
k), which may have been due to the southward delay of the northerly air flow.

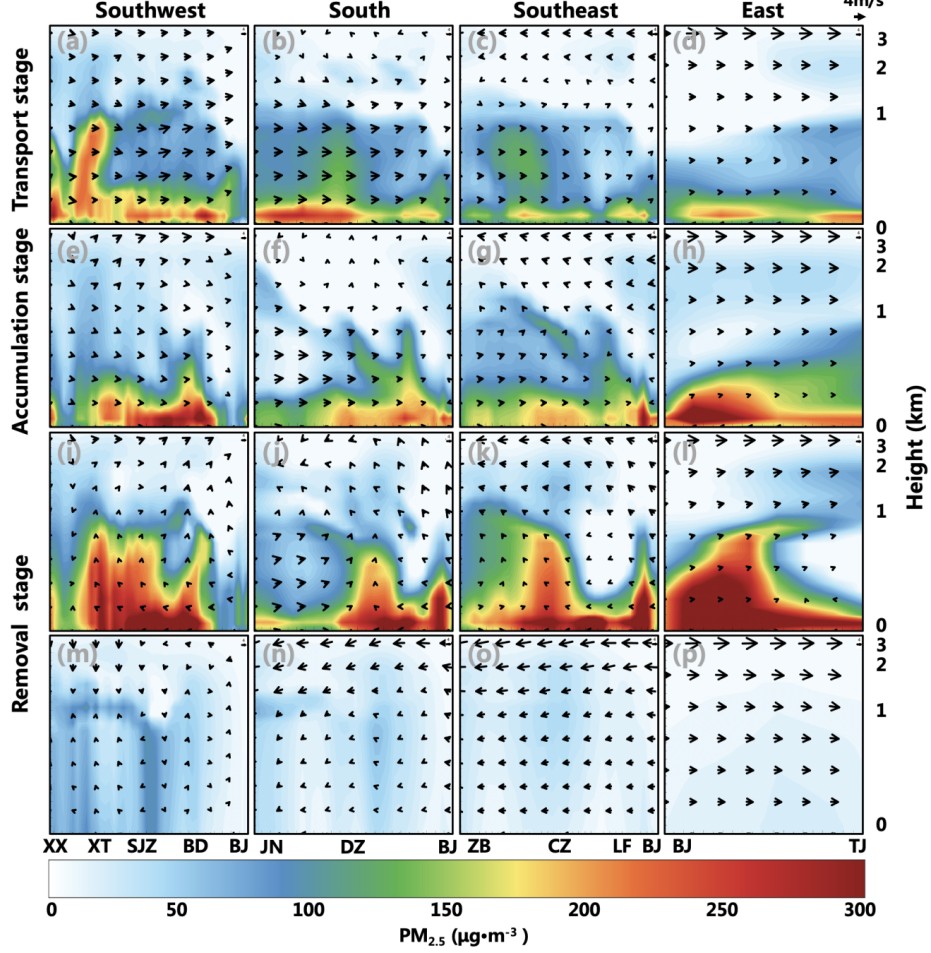

**Figure 8**. Vertical profiles of PM₂.₅ cross-sections with wind vectors along the transport pathways,
including southwest (first column), south (second column), southeast (third column), and east (fourth
column). The first row (00:00 December 2, 2017) represents the transport stage, the second row (10:00
December 2, 2017) represents the accumulation stage, the third row (00:00, December 3, 2017) and the
fourth row (14:00, December 4, 2017) represent the removal stage.
To investigate the vertical variation of PM$_{2.5}$ inflow or outflow at different heights and
determine the height at which the main transport occurred (Zhang et al., 2019b), we plotted the
vertical distribution of PM$_{2.5}$ transport flux in different directions (Fig. 9). Here the PM$_{2.5}$
transport flux is defined as the product of PM$_{2.5}$ mass concentration (µg m$^{-3}$), wind speed (m s$^{-1}$),
), and wind direction projection in the current pathway (Xiang et al., 2020). The southwest,
southeast, and east pathways in Fig. 9 were represented by BD, LF, and TJ, respectively, which
are the three lidar stations closest to BJ (Fig. 1). TF > 0 indicates that the pollutants were
imported to Beijing, while TF < 0 indicates that the pollutants were exported from Beijing. The
results revealed that below the height of 1.5 km, the order of the maximum values of imported
pollutants to Beijing direction was southwest pathway (1122.8 µg m$^{-2}$ s$^{-1}$) > southeast pathway
(423.6 µg m$^{-2}$ s$^{-1}$) > east pathway (278.3 µg m$^{-2}$ s$^{-1}$), while the exported pollutants from Beijing
direction was southwest pathway (-1571.4 µg m$^{-2}$ s$^{-1}$) > east pathway (-877.7 µg m$^{-2}$ s$^{-1}$)>
southeast pathway (-772.4 µg m$^{-2}$ s$^{-1}$). Compared with the PM$_{2.5}$ transport flux on the ground
surface, the relatively high value (~ 200 µg m$^{-2}$ s$^{-1}$) in the southwest pathway (Fig. 9a) occurred
on November 29 and early morning on December 4, while the relatively extreme value (~ -400
µg m$^{-2}$ s$^{-1}$) on the east pathway (Fig. 9c) was recorded at the night of December 2.

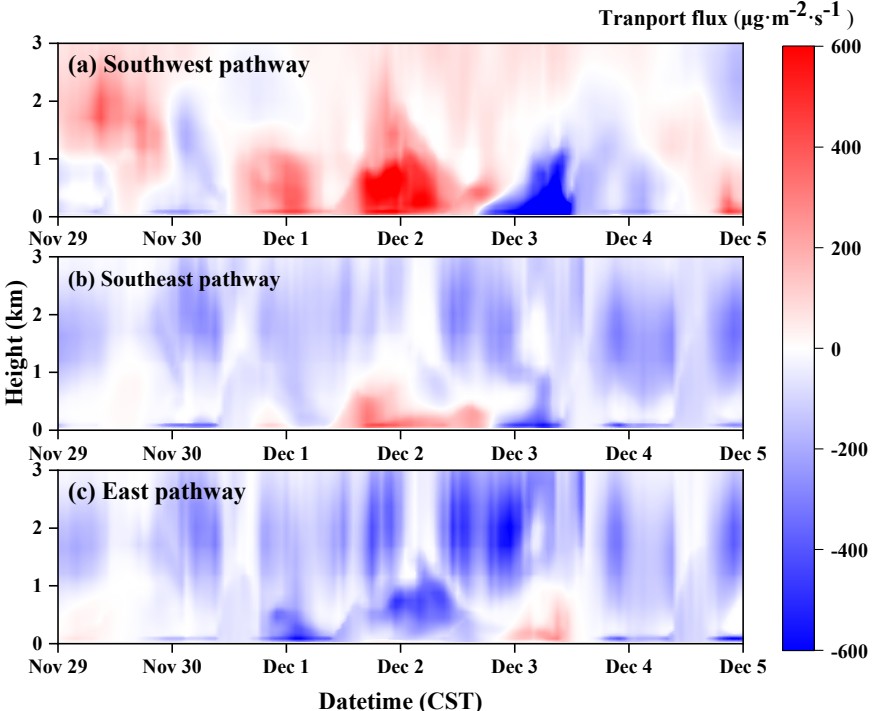

**Figure 9.** Time series of PM$_{2.5}$ transport flux from different transport pathways. The corresponding
directions of the southwest, southeast, and east pathways are shown in Fig. 1.



To further obtain insights into the total transport characteristics in the target area (BJ)
and its surrounding area (BD, LF, and TJ) during different evolutionary stages, the time series
of the PM$_{2.5}$ transport flux intensity (TFI) was shown in Fig. 10, which was calculated by
integrating the PM$_{2.5}$ transport flux within the height range of 1.5 km. The TFI of PM$_{2.5}$ further
reveals that pollutants imported into the Beijing area with a maximum PM$_{2.5}$ TFI of ~4.6×10$^5$
μg·m$^{-1}$·s$^{-1}$ were transported mainly via the southwest pathway during the TP, while the extreme
TFI of pollutants exported from Beijing via the east pathway was approximately -1.4×10$^5$
μg·m$^{-1}$·s$^{-1}$. In addition, during the RP, the pollutants from Beijing were exported to the
southwest and southeast, with extreme values of approximately -1.03×10$^6$ and -4.3×10$^5$ μg·m$^{-}$
$^{1}$·s$^{-1}$, respectively. On the contrary, the absolute value of TFI on the southwest pathway was <
~1.0×10$^4$ μg·m$^{-1}$·s$^{-1}$ during the EP (Fig. 10), which indicates that there was no significant
inflow or outflow of pollutants. However, this reason was mainly due to the offsetting of the
inflow of pollutants in the upper-air and the outflow of pollutants near the ground (Fig. 9a).
This special phenomenon also demonstrates that the study of vertical distribution of pollutants
has great significance, which can better explain the transport characteristics (Zhang et al.,
2019b).

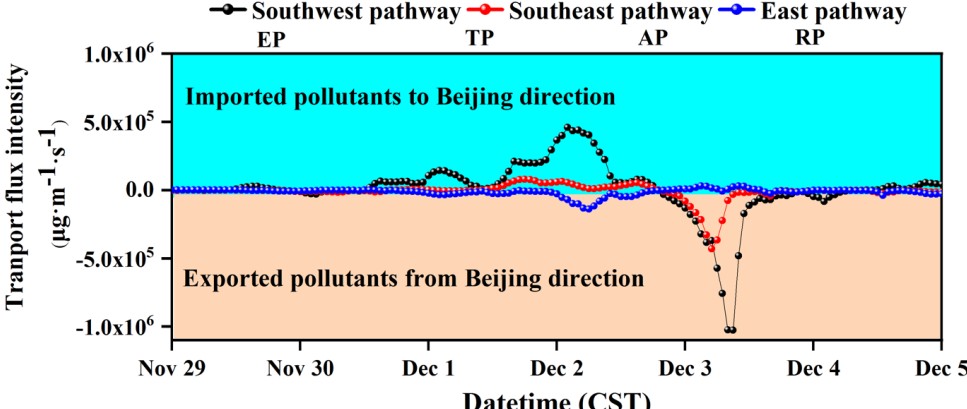

**Figure 10.** Time series of PM$_{2.5}$ transport flux intensity from different transport pathways. The
corresponding directions of the southwest, southeast, and east pathways are shown in Fig. 1.
**4   Conclusions**
Accurate quantification of the distribution of particulate matter in the atmosphere is a
key requirement for predicting air quality and estimating atmospheric environmental capacity
from atmospheric observations. We utilized a vertical observation network composed of 13
aerosol lidars, combined with data assimilation technology, to improve the simulation accuracy



of $PM_{2.5}$, and further analyzed the multi-dimensional evolutionary characteristics of pollutants
in the surface layer, vertical layer, and 3-D space, thereby providing effective data support for
clarifying the spatial transport characteristics of heavy pollution.

4       We found that the average height of the atmospheric boundary layer was < 0.5 km

during the HAP period. We further demonstrated that the transport of pollutants in the NCP
region was mainly via three pathways: southwest, southeast, and east. During the TP, the $PM_{2.5}$
advected into Beijing with a maximum transport flux intensity (TFI) of ~$4.6\times10^5$ µg·m$^{-1}$·s$^{-1}$
was mainly via the southwest pathway, while the polluted air mass in the RP dissipated from
Beijing via the southwest and southeast pathways, with extreme $PM_{2.5}$ TFI values of
approximately $-1.03\times10^6$ and $-4.3\times10^5$ µg·m$^{-1}$·s$^{-1}$, respectively. In addition, the transport of
regional pollutants to the Yangtze River Delta was due to the continuous southward flow of
northwest and northeast winds. Our results directly revealed that pollutants in the North China
Plain can be transported to the Yellow Sea and the Bohai Sea, providing a dataset for a further
in-depth study of the mechanism of air pollution in the coastal areas of eastern China. This
study also captured the regional transport of air pollutants stretching over 1000 km, proving
the necessity and importance of the joint prevention and control of regional air pollution.

## Data availability

18       The FNL data are available from the following website

(https://rda.ucar.edu/datasets/ds083.2/). The data in this study are analyzed using the NCAR
Command Language (http://www.ncl.ucar.edu/). The authors are gratefully acknowledging the
China National Environmental Monitoring Center for providing monitoring data for the $PM_{2.5}$
(http://106.37.208.233:20035). The lidar data in this study are available upon request from the
corresponding author (yxiang@ahu.edu.cn).

## Author contributions

YX and TZ designed this study. YX wrote the manuscript; YC and CM edited it. LL

and TZ helped to analyze the data. YC, CM, WL, and JL provided constructive comments on
this study. All authors contributed to the discussion and final version of the manuscript.

## Competing interests

29       The authors declare that they have no conflict of interest.

## Acknowledgements

31       This work was supported by the National Natural Science Foundation of China

32   (42005106, 41941011), and the National Key Project of MOST (2017YFC0213002,



2018YFC0213101, 2018YFC0213106, 2018YFC0213201), and the Major science and technology projects of Anhui Province (No.18030801111), and the Natural Science Foundation of Anhui Province, China (1908085QD160, 1908085QD170), and the Doctoral Scientific Research Foundation of Anhui University (Y040418190). The authors are grateful to the China National Environmental Monitoring Center for providing the $PM_{2.5}$ monitoring data. The authors also gratefully acknowledge © Google Earth for providing the map used in this research. Yafang Cheng and Chaoqun Ma thank the Minerva program of Max Planck Society.

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
