# Peer review of "Lidar vertical observation network and data assimilation reveal key processes driving the 3-D dynamic evolution of PM2.5 concentrations over the North China Plain"

_Atmospheric Chemistry and Physics, 2020_

## Author Comment (AC1)

**Response to the Comments of Referee #1**

**Lidar vertical observation network and data assimilation reveal key processes driving the 3-D dynamic evolution of PM$_{2.5}$ concentrations over the North China Plain**

Yan Xiang, Tianshu Zhang, Chaoqun Ma, Lihui Lv, Jianguo Liu, Wenqing Liu, and Yafang Cheng

We appreciate the reviewer's comments on the manuscript. All comments are highly valuable and helpful for us to improve our manuscript. We have studied them carefully and have addressed them in the revised manuscript. Below we address the reviewers' comments, with the reviewer comments in black, and our response in blue. We have revised the manuscript accordingly, and mentioned the line number of the tracked revision.

**Anonymous Referee #1:**

**Summary and general comments:**

This manuscript reports the evolution characteristics of PM$_{2.5}$ concentration in different dimensions (surface-layer, vertical-distribution and three-dimensional) under four different phases (an early phase, a transport phase, an accumulation phase, and a removal phase) of heavy pollution process in North China Plain. The authors used data from an observation network consisting of 13 aerosol lidars and more than 1000 ground observation stations, combined with a data assimilation technique, to conduct a comprehensive analysis of an extreme heavy aerosol pollution over the North China Plain from November–December 2017. Meanwhile, the regional transport of PM$_{2.5}$ over different transport channels was quantified, including PM$_{2.5}$concentration, transport flux and transport flux intensity. Moreover, the authors also captured the regional transport of air pollutants stretching over 1000 km, proving the necessity and importance of the joint prevention and control of regional air pollution.

These results could significantly improve our understanding on the key processes driving the 3-D dynamic evolution of $PM_{2.5}$ concentrations. The scope of this manuscript is well suited to ACP, and the data obtained by the authors are valuable. The data set is meaningful to further verify or constrain the representation of aerosols in air quality model and satellite remote sensing. This paper is very well-written and should be considered for publication after addressing my comments below.

**Thank you very much for your encouraging comments.**

**List of minor comments:**

1. Page 4, Line 10: The map information shown in Fig. 1a and Fig. 1b is too duplicate with that shown in Fig. 1c and Fig. 1d respectively. It is suggested to delete Fig. 1c and Fig. 1d or put them in the supporting material.

   **Thanks for the suggestions; we have moved figures 1c and 1d into the supporting material. Please see line 12 on page 4 in the revised manuscript and Figure S1.**

2. Page 5, Line 5: The time resolution of 3-10 minutes refers to the time resolution of the original data or the smoothed data. If it is original, please describe clearly; if it is smooth, please give a detailed smoothing method in the manuscript.

   **Yes, the time resolution here refers to the original data, that is, the interval time of each profile. We have added a clear description to the revised manuscript. See line 21 on page 4.**

3. Page 5, Line 6: The semicolon should be changed to a comma.

   **Corrected, please see line 22 on page 4 in the revised manuscript.**

4. Page 5, Line 17: Please provide the WRF Chem version used in the manuscript.

   **Thanks. We have provided the version information of WRF-Chem (version 3.8.1) in the revised manuscript. Please see line 9 on page 5 in the revised manuscript.**

5. Page 6, Line 26: A space needs to be added between 1 and km to be consistent with other contents of the manuscript.

**Corrected, please see line 18 on page 6 in the revised manuscript.**

6. Page 9, Line 7: Xintai should be Xingtai.

**Thanks. They were changed, please see line 3 on page 9 in the revised manuscript.**

7. Page 12, Line 5: What is the meaning of white color in j, k and l of Figure 6? Does it mean that the current moment is missing data? Or is it deleted due to low SNR? Please add a clear description to the manuscript.

**Yes, the reason for the white color in Figure 6 is that these lidars were not working normally at the current time, resulting in the missing of data. We have added an explanation to the revised manuscript. Please see line 11 on page 12.**

8. Page 16, Line 4: Why choose 1.5 km to calculate the total amount of $PM_5$ transportation. Why not 1 km or 2 km, 3 km? Please give reasons. Is it based on the height of the atmospheric boundary layer? Or is it due to the 1.5 km explained in line 9 on page 15?

**Thanks for the suggestion. Most aerosol pollutants were centralized near the surface, while a part of particles can also be transported to the height of 1–3 km from the ground (Figure 6). Therefore, this work focuses on the horizontal transport of $PM_{2.5}$ within a height of 3 km (Figure 7 & 8 & 9).**

**In addition, the vertical profiles of $PM_{2.5}$ cross-sections on different transport channels reveal that the pollutant transport mainly occurs within 1.5 km (Figure 8), which is also mainly the height of the boundary layer (Figure 7). Therefore, the $PM_{2.5}$ transport flux intensity (TFI) was calculated up to a height of 1.5 km**

**We have added the details in the revised version; please see lines 3-6 on page 16.**

---

## Author Comment (AC2)

**Response to the Comments of Referee #2**

**Lidar vertical observation network and data assimilation reveal key processes driving the 3-D dynamic evolution of PM$_{2.5}$ concentrations over the North China Plain**

Yan Xiang, Tianshu Zhang, Chaoqun Ma, Lihui Lv, Jianguo Liu, Wenqing Liu, and Yafang Cheng

We appreciate the reviewer's comments on the manuscript. All comments are highly valuable and helpful for us to improve our manuscript. We have studied them carefully and have addressed them in the revised manuscript. Below we address the reviewers' comments, with the reviewer comments in black, and our response in blue. We have revised the manuscript accordingly, and mentioned the line number of the tracked revision.

**Anonymous Referee #2:**

**General Comments:**

Xiang et al. report on using three-dimensional variational data assimilation to refine WRF-Chem simulations of PM2.5 transport throughout the North China Plain based on surface and lidar observations. This paper extends on a number of other recent studies from this region by incorporating aerosol vertical profiles from a network of 13 lidars located along the main corridors for air pollution transport. The resulting three-dimensional characterisation of PM2.5 concentrations and fluxes allows characterisation of the inflow and outflow pathways for this region and the vertical structure of heavy aerosol pollution events. Furthermore, the authors were able to identify altitude-dependent differences in flux rates and direction.

The manuscript is well written and within the scope of Atmospheric Chemistry and Physics. While only examining a single heavy aerosol pollution event, the method may significantly enhance aerosol transport models in this region and could be particularly

valuable in assessing air pollution control strategies. The study is presented in a clear and engaging manner and should be considered for publication after addressing the following minor comments:

**Thank you very much for your encouraging comments.**

**Specific Comments:**

Page 6, Line 33: Are the quoted root-mean-square errors and correlation coefficients calculated from the combined data at the four selected heights (surface, 0.2, 0.5 and 1 km)?

**Yes, the root-mean-square errors and correlation coefficients are calculated based on the combined data at the four selected heights (ground, 0.2, 0.5 and 1 km). We have added a clear statement to the revised manuscript, see line 24 on page 6.**

Page 11, Line 14: Although elevated concentrations are briefly visible at approximately 1km over HD and XX in the removal phase (Figs 6f & 6g), it is not immediately clear that this corresponds to north-south transport from BJ. Perhaps some elaboration is required or at least the upward wind vectors shown in Fig 8 could be mentioned here.

**I'm sorry that we didn't describe it clearly here. Actually, these pollutants in the upper air come from the local emissions on the ground, which is due to the updraft lifting to 1-2 km above the ground on the night of November 28.**

**We have added the above discussions and figures (Fig. S4) in the revised version. See lines 13-15 on page 11.**

[Figure]

**Figure S4.** Time series of vertical distributions of the aerosol extinction coefficient (first column) observed and vertical wind velocity (second column) simulated on the North China Plain from November 28–November 30, 2017. Missing datasets are plotted in white.

Page 15, Line 9: As suggested by the other reviewer, some reasoning should be included to explain why the TFI was calculated up to a height of 1.5 km, rather than some other limit.

**Thanks for the suggestion. Most aerosol pollutants were centralized near the surface, while a part of particles can also be transported to the height of 1–3 km from the ground (Figure 6). Therefore, this work focuses on the horizontal transport of PM$_{2.5}$ within a height of 3 km (Figure 7 & 8 & 9).**

**In addition, the vertical profiles of PM$_{2.5}$ cross-sections on different transport channels reveal that the pollutant transport mainly occurs within 1.5 km (Figure 8), which is also mainly the height of the boundary layer (Figure 7). Therefore, the PM$_{2.5}$ transport flux intensity (TFI) was calculated up to a height of 1.5 km**

**We have added the details in the revised version, see lines 3-6 on page 16.**

**Additional Comments:**

Figure 3: For clarity, the episode numbers should be centered over each episode

**Corrected, see Figure 3 in the revised manuscript.**

Figures 4 & 5: Figure 5 should be inserted before Figure 4 since it is discussed first in the text (page 10)

**Thank you for your advice. In fact, figure 4 is described in line 13 on page 9 (Section 3.2), while Figure 5 is described in Section 3.3.**

Page 13, line 11: Change "come from" to "coming from"

**Corrected, see line 14 on page 13 in the revised manuscript.**

Page 15, line 13: It is not clear what is being compared against the ground surface flux. What are the height of these fluxes? Or is this sentence providing ground surface fluxes for comparison against the maximum values across the 0 – 1.5 km range, as given in the previous sentence?

**Thank you for this comment. We agree with the reviewers that the description of this sentence is not very clear. In order to avoid confusion, we have deleted it in the revised manuscript, which will not affect the conclusion of the paper.**

Page 16, line 10: "On the contrary" implies that the TFI value for EP contradicts the value for RP. Perhaps "In contrast" or "In comparison" would be more appropriate.

**Thanks. According to your suggestion, we have reworded this sentence. See line 11 on page 16 in the revised manuscript.**